# PEGylated Nanographene Oxide in Combination with Near-Infrared Laser Irradiation as a Smart Nanocarrier in Colon Cancer Targeted Therapy

**DOI:** 10.3390/pharmaceutics13030424

**Published:** 2021-03-22

**Authors:** Milena Georgieva, Zlatina Gospodinova, Milena Keremidarska-Markova, Trayana Kamenska, Galina Gencheva, Natalia Krasteva

**Affiliations:** 1Institute of Molecular Biology “R. Tsanev”, Bulgarian Academy of Sciences, “Acad. Georgi Bonchev” Str., Bl. 21, 1113 Sofia, Bulgaria; 2Institute of Biophysics and Biomedical Engineering, Bulgarian Academy of Sciences, “Acad. Georgi Bonchev” Str., Bl. 21, 1113 Sofia, Bulgaria; zlatina.go@abv.bg (Z.G.); m_keremidarska@abv.bg (M.K.-M.); trayanakamenska@abv.bg (T.K.); 3Institute of Plant Physiology and Genetics, Bulgarian Academy of Sciences, “Acad. Georgi Bonchev” Str., Bl. 21, 1113 Sofia, Bulgaria; 4Faculty of Biology, Sofia University “St. Kliment Ohridski”, 8 Dragan Tsankov Blvd, 1164 Sofia, Bulgaria; 5Faculty of Chemistry and Pharmacy, Sofia University “St. Kliment Ohridski”, 1 James Bourchier Blvd., 1164 Sofia, Bulgaria; gkissovsky@gmail.com

**Keywords:** nano-graphene oxide (nGO), nGO-PEG, near-infrared (NIR) light, phototherapy, colon cancer

## Abstract

Anti-cancer therapies that integrate smart nanomaterials are the focus of cancer research in recent years. Here, we present our results with PEGylated nanographene oxide particles (nGO-PEG) and have studied their combined effect with near-infrared (NIR) irradiation on low and high invasive colorectal carcinoma cells. The aim is to develop nGO-PEG as a smart nanocarrier for colon cancer-targeted therapy. For this purpose, nGO-PEG nanoparticles’ size, zeta potential, surface morphology, dispersion stability, aggregation, and sterility were determined and compared with pristine nGO nanoparticles (NPs). Our results show that PEGylation increased the particle sizes from 256.7 nm (pristine nGO) to 324.6 nm (nGO-PEG), the zeta potential from −32.9 to −21.6 mV, and wrinkled the surface of the nanosheets. Furthermore, nGO-PEG exhibited higher absorbance in the NIR region, as compared to unmodified nGO. PEGylated nGO demonstrated enhanced stability in aqueous solution, improved dispensability in the culture medium, containing 10% fetal bovine serum (FBS) and amended biocompatibility. A strong synergic effect of nGO-PEG activated with NIR irradiation for 5 min (1.5 W/cm^−2^ laser) was observed on cell growth inhibition of low invasive colon cancer cells (HT29) and their wound closure ability while the effect of NIR on cellular morphology was relatively weak. Our results show that PEGylation of nGO combined with NIR irradiation holds the potential for a biocompatible smart nanocarrier in colon cancer cells with enhanced physicochemical properties and higher biological compatibility. For that reason, further optimization of the irradiation process and detailed screening of nGO-PEG in combination with NIR and chemotherapeutics on the fate of the colon cancer cells is a prerequisite for highly efficient combined nanothermal and photothermal therapy for colon cancer.

## 1. Introduction

Colon cancers, also designated as colorectal cancers (CRCs), are the third most commonly diagnosed form of cancer globally, comprising 11% of all cancer diagnoses with about 1.8 million new cases registered in 2018 [1]. The CRC is more incident among men than women and three to four times more common in developed than in developing countries. In Western countries, colon cancer is primarily a disease of the elderly, with a peak incidence at around 70 years [2]. The burden, however, is shifting to younger individuals according to the Colorectal Cancer Statistics 2020 [3]. In 2015 and 2016, the median age at diagnosis was 66 years, compared with 72 years in 2001 and 2002. So far, the conservative therapeutic approaches like radical surgery to remove the tumor mass remain the primary treatment, even though, in advanced stages, this appears ineffective [1,3,4,5]. Exactly in the advanced stages of colon cancer, the patients undergo chemotherapy, which, in many cases, is also futile, as often the tumor cells develop drug resistance [6,7]. For this reason, oncological research is investing many efforts toward finding new and efficient therapies that can overcome the limitations of conventional treatments [8,9,10]. 

Nanomaterials offer numerous attractive physical, chemical, optical, and electronic properties, such as light, thermal-responsiveness, and pH-responsiveness, which, together with their nano dimensions and relatively good biocompatibility, make them appropriate in drug delivery systems and photothermal therapy (PTT) for cancer [11,12,13]. PTT involves strong light absorption by near-infrared (NIR) responsive nanomaterials including carbon nanomaterials, with subsequent energy dissipation as heat [14,15]. Furthermore, nanomaterials act as photothermal absorbers and enhance PTT selectivity within the target tumour tissue. The last is achieved within therapeutic temperatures above 41°C, which, due to their little total light energy, minimize the damage to the healthy surrounding tissues. Since nanomaterials function as nanocarriers for such molecular photosensitizers or chemotherapeutic drugs, they are also able to integrate chemotherapy and PTT treatment modalities into a single, smart nanocarrier, thus, leading to advanced, personalized cancer treatment with superior therapeutic performance and minimal side effects [16]. 

Graphene derivatives have attracted recently tremendous interest for application in novel approaches for oncological practice. The reason for this is the functional and chemical characteristics of graphene. Graphene is a single-layered material composed of sp^2^-hybridized carbon atoms arranged in a honeycomb network [17]. Its two-dimensional structure provides a highly specific surface area for the binding of a high number of different molecules including drugs, cell-targeting ligands, nucleic acid, and proteins efficiently [18,19,20]. Among the variety of graphene derivates, graphene oxide (GO) and graphene quantum dots (GQDs) are the most extensively explored for light-mediated approaches in cancer treatment, namely photothermal and photodynamic therapies due to their strong visible and near-field absorbance and low toxicity of GQDs [20,21,22]. Additionally, GO nanoparticles not only function as effective drug carriers, but also have the potential to exert adverse effects on cancer cells by inducing the generation of reactive oxygen species (ROS) and DNA damage in cancer cells [23]. Our previous expertise with pristine and aminated GO nanosheets demonstrated that amination enhanced the GO therapeutic effects in colorectal and hepatocellular carcinoma cells [24,25,26,27]. Nanoparticles (NPs)-mediated ROS production is considered as the primary source of nanoparticles’ toxicity [28]. The high levels of ROS may cause nucleic acids oxidation, lipid peroxidation, mitochondrial damage, and DNA strand-breaks, which result in cell death and may trigger inflammation and fibrosis. However, because the potential anticancer drugs have to be maximally effective in cancer cells with a minimum damage of normal cells to avoid the toxicity of the surrounding healthy tissue, GO NPs need to be improved for their biocompatibility. Graphene-induced toxicity arises from the complex interplay between particle characteristics (e.g., size, shape, surface chemistry and charge, solubility, dispersion, aggregation), particle concentration, and exposure time and cell type. Many studies reported the propensity of GO to form aggregates, particularly under physiological conditions. For instance, fast sedimentation and aggregates formation of graphene nanoparticles inhibit the nutrient uptake of human skin fibroblast cells [29]. Thus, the aggregations and sedimentation of graphene particles compromise their biocompatibility. One approach to stabilizing GO NPs in aqueous dispersions is to coat their surface with polymeric ligands that provide the NPs surface with a physical barrier [30]. Polyethylene glycol (PEG) is widely used to functionalize different nanomaterials due to its hydrophilic properties [31,32,33,34,35,36]. PEGylation is the process in which graphene NPs are functionalized with PEG. The last has been reported to enhance nanoparticles’ biocompatibility and to improve their chemical properties. Some of these PEGylated NPs are used as nano-carriers in cancer therapies [37,38]. Moreover, PEG has low cytotoxicity, good water solubility, and stability. It has been proven that PEG preserves nanoparticles from aggregation, opsonisation, and phagocytosis [39,40].

Based on the above considerations, herein, we provide further insight into the potential of PEGylated nGO as a perspective water-soluble nano-formulation as an optimized nanocarrier for colon cancer treatment. We have functionalized nanoscale graphene oxide (nGO) sheets with PEG to improve solubility and stability in physiological solutions and have evaluated the biological compatibility of nGO and nGO-PEG alone and in combination with near-infrared NIR laser radiation. Our results demonstrated that PEGylation improves stability, aggregation, and biocompatibility of nGO, thus, making it a suitable molecule for targeted colon cancer therapy. Furthermore, the combination of GO-PEG with near-infrared light proved to be more biologically compatible, preserving cells viability and morphology. nGO-PEG with NIR inhibited the cell migration of low invasive (HT29) colon cancer cells, thus, paving the road for further experiments. Though promising, these results highlight the need for further optimization of the irradiation process in synergy with nGO-PEG therapy to develop highly efficient combined nano and thermal therapy for colorectal carcinoma cells.

## 2. Materials and Methods

### 2.1. Preparation of Poly(ethylene glycol) Modified Reduced Graphene Oxide (nGO-PEG)

PEGylation of pristine nGO was performed using a previously established method [41]. In brief, pristine GO (4 mg/mL, Graphenea, Spain) was diluted in deionized water to a concentration of 1 mg/mL and sonicated for 2 h at 500 W on an ultrasonic homogenizer (VCX 500, Sonics and Materials, Inc., Newtown, CT, USA) to obtain nanosized graphene oxide (nGO). At the end of sonication, mPEG-NH_2_ (Abbexa Ltd., Cambridge, UK), was added and further sonicated for 5 min. The suspension was allowed to react overnight at 70 °C in a water bath. The resulting nGO-PEG suspension was centrifuged at 6000× *g* for 10 min to remove any unstable aggregates and stored at 4 °C until further use. For all biological studies, just before exposure to cells, the NPs solutions with a concentration of 1.0 mg/mL were sonicated for 1 h in a water bath (UM-2, Unitra). After that, NPs were added to the cells in different concentrations.

### 2.2. Physicochemical Characterization of nGO and nGO-PEG

#### 2.2.1. Fourier Transform Infrared (FTIR) Spectroscopy

Infrared spectra of nGO and nGO-PEG dehydrated samples were recorded using a Thermo Nicolet 6700 spectrometer (Thermo Fisher Scientific, Waltham, MA, USA) in the mid-infrared region (4000–400 cm^−1^) at a resolution of 2 cm^−1^. 

#### 2.2.2. UV-Visible Spectroscopy

UV–VIS-absorption spectra were acquired using a Specord 210 Plus (Edition 2010, Analytik Jena AG, Jena, Germany) spectrophotometer. Samples were transferred to a 200-µl quartz cuvette and spectra were recorded in the range of 190–1100 nm.

#### 2.2.3. Raman Spectroscopy

The Raman spectra were obtained using LabRAM HR Visible Raman spectrometer (HORIBA Jobin Yvon, Villeneuve d’Ascq, France). For excitation, the 633 nm line of a He-Ne laser was used. To decrease the possible local overheating, the used laser power on the sample surface was reduced to 10 μW. An objective ×50 was used both to focus the incident laser light and to collect the scattered light in backscattering geometry. Several spectra from different arbitrary spots from each sample were collected. The investigated spectral range was 1000 cm^−1^ to 4000 cm^−1^.

#### 2.2.4. Transmission Electron Microscopy

The morphology of nGO and nGO-PEG was analyzed by a transmission electron microscope, TEM (JEM-2100, Tokyo, Japan). Samples were resuspended in deionized water and sonicated for 60 min in a water bath. A drop of NPs suspensions was placed on a carbon grid and the grid was completely dried and observed under TEM at 200 KeV.

#### 2.2.5. Particle Size Distributions, Zeta Potential, and Polydispersity Index (PDI) Measurements

Particles size distributions’ zeta potential and PDI for nGO and nGO-PEG were measured at 25 °C by the dynamic light scattering (DLS) method, using a Zetasizer (Malvern Instrument, Ltd., Worcestershire, UK). Before measurements, nGO and nGO-PEG dispersions with concentrations of 1 mg/mL were sonicated in a water bath for 60 min. The zeta potential of the aqueous dispersions was determined in a disposable Zetasizer cuvette. Each measurement was performed in triplicate at room temperature and results are reported as MEAN ± standard deviation (STDV).

### 2.3. Colloidal Stability and Aggregation Behaviour of nGO and nGO-PEG Dispersion in Aqueous Solution

To evaluate the colloidal stability of nGO and nGO-PEG in aqueous solution, as-prepared NPs dispersions with a concentration of 1 mg/mL were measured on the Malvern apparatus to determine the alteration in their size and zeta potential under 10 days storage at 4 °C. The measurements were done on the first and tenth days at 22 °C. All values reported were determined from the MEAN values of three successive measurements (MEAN ± STDV). For evaluation of aggregation behaviour of nGO and nGO-PEG NPs, graphene derivatives (100 µg/mL) were incubated with cells and Dulbecco’s modified Eagle’s medium (DMEM) medium containing 10% FBS (fetal bovine serum) for five days and photos were taken by Nikon D700 camera on first and fifth days. 

### 2.4. Sterility of nGO and nGO-PEG

Pristine GO and nGO-PEG nanosheets were streaked on three different types of agar plates, including Columbia agar with 5% Sheep Blood (Becton Dickinson) for Gram (+) and Gram (–) microorganisms, MacConkey agar (Becton Dickinson)—a selective and differentiating agar for gram-negative bacterial species, and Sabouraud Dextrose Agar (Becton Dickinson)—a nonselective medium for pathogenic and non-pathogenic fungi, particularly dermatophytes. *Staphylococcus aureus*, *Escherichia coli*, and *Candida albicans* were used as positive controls of bacterial and fungal growth. Agar plates were incubated for 72 h at 37 °C. Microbial and fungal cell growth was evaluated and results are photographed with a Nikon D700 reflex camera (Nikon, Tokyo, Japan).

### 2.5. In Vitro Biological Studies on High and Low Invasive Colon Cancer Cells

#### 2.5.1. Cell Culture 

Biological studies involved two different types of cell cultures: Colon 26: a mouse highly invasive colon adenocarcinoma cell line (ATCC, CRL-2638) and HT29: a human low invasive colon adenocarcinoma cell line (ATCC, HTB-38). Both cell lines were purchased from FOT, Ltd., Sofia, Bulgaria. Cells were cultured in Dulbecco’s modified Eagle’s medium (DMEM) supplemented with 10% (*v*/*v*) fetal bovine serum (Sigma-Aldrich, Germany) and 1% (*v*/*v*) penicillin/streptomycin (Sigma-Aldrich, Germany) in a humidified atmosphere with 5%CO_2_/95% air at 37 °C. For cell experiments, the confluent cells were harvested and seeded in a concentration depending on the type of assay described below (1 × 10^4^–4 × 10^5^ cells/mL). Then the cells were incubated for 24 h before exposure to increasing concentrations of NPs.

#### 2.5.2. Cell Viability and Proliferation Assays

The effect of nGO and nGO-PEG without NIR irradiation on cell viability and proliferation was evaluated using different concentrations in a range between 5 and 500 µg/mL. Cells were seeded in 96-well plates at a density of 1 × 10^4^ cells/well and incubated at 37 °C and 5% CO_2_. After 24 h of incubation, the culture medium was renewed. nGO or nGO-PEG NPs at different concentrations were added to a final volume of 200 µl/well (in complete DMEM) and cells were incubated with the materials for 24 h and 72 h. Cell viability and growth were then quantified by the WST-1 assay (2-(4-iodophenyl)-3-(4-nitrophenyl)-5-(2,4-disulfophenyl)-2H-tetrazolium). Briefly, the cell culture medium containing NPs was removed. Cells were washed with Phosphate Buffered Saline (PBS) and incubated in 10% (*v*/*v*) WST-1 reagent (Sigma-Aldrich, St. Louis, MO, USA) in culture medium at 37 °C and 5% CO_2_ for 2 h. The amount of formazan was measured by absorbance at 450 nm using a standard microplate reader spectrophotometer (Thermo Scientific Multiskan Spectrum). Control cells were incubated with DMEM without NPs. Data for each sample were normalized to the control and results are presented as a percentage of control viability. All assays were performed in triplicates and the estimated values are presented as MEAN ± STDV.

#### 2.5.3. Photothermal Near-Infrared Irradiation Assays

To evaluate the combined effect of nGO and nGO-PEG NPs and NIR irradiation, 5 × 10^4^ cell/mL Colon26 and HT29 cells were seeded into 24-well plates and incubated with 100 µg/mL NPs as described above. After 24 h of incubation with nGO and nGO-PEG NPs, cells were irradiated for 5 min at room temperature using a NIR-based source (laser) with peak emission around 808 nm (NIR region) and irradiance of 1.5 W/cm^2^. For long-term assays (72 and 96 h), the cells were NIR irradiated every day. Immediately before the assays, the medium containing nGO and nGO-PEG dispersions was removed. The cells were washed with PBS, and all assays for biological activity were performed as described. To compare the effects of NIR irradiation, negative control in the absence of the tested nanomaterials and without NIR irradiation was used for normalization. 

#### 2.5.4. Cell Morphology

Vital morphology of Colon26 and HT29 cells treated with nGO and nGO-PEG NPs and NIR irradiated was visualized by Fluorescein Diacetate (FDA) staining [24]. Cells were plated in 24-well plates containing sterile cover glasses, treated with NPs, and NIR –irradiated, as described above. At the end of incubation, FDA (final concentration of 1 μgmL^−1^) was added into the culture plates and the cells were incubated for 2 min at room temperature. The living cells were stained green by FDA, and observed under an inverted fluorescence microscope (Axiovert 25, Carl Zeiss, Germany), equipped with a CCD camera. Untreated cells (without nGO and nGO-PEG NPs and non-irradiated cells) were used for normalization. 

#### 2.5.5. Cell Migration Assays 

To assess the effect of nGO and nGO-PEG NPs on cell migration, an in vitro wound healing experiment was performed. Colon26 and HT29 cells were seeded in 6-well plates with a concentration of 4 × 10^5^ cells/well and cultured at 37 °C and 5% CO_2_ to 90–95% confluence. A wound scratch was performed to disrupt the confluent cell monolayer. Cells were rinsed with PBS to remove cell debris and fresh medium, containing 1% FBS and 100 µg/mL of both types of NPs was added while the cells were further incubated for 96 h. In addition, 24 hours after NPs tumour treatment, the cells were irradiated by 808 nm laser with a power density of 1.5 W/cm^2^ for 15 min. Once a day for five consecutive days, the cells were NIR irradiated for 15 min and the cell migration was observed by phase-contrast microscopy for four days, which was sufficient to complete the scratch closure in the control cells (without nanoparticles and NIR). The area devoid of cells was calculated in pixels on days 1, 3, and 5 by Image J. The percentage of the scratched area in each group of cells at these three-time points was calculated using the following formula.
% wound closure = (width of the scratch wound at 0 h − width of the scratch wound at 24/72 or 120 h)/width of the scratch wound at 0 h × 100

### 2.6. Statistical Analysis 

Data were statistically analyzed by Student’s-test and one-way analysis of variance (ANOVA) followed by Dunnett’s post-hoc test and the data are considered significant when *p* < 0.05, designated with *.

## 3. Results

### 3.1. Physicochemical Characteristics of Nanoparticles

#### 3.1.1. Spectroscopic Characterization of nGO and nGO-PEG by Fourier Transform Infrared (FTIR) and UV-Visible Spectroscopy

Graphene oxide (GO) and PEGylated GO suspensions were obtained in water and culture medium and the first steps in our experiments were their physicochemical characterization (Figure 1). The Fourier Transform Infrared (FTIR) spectroscopy gave simultaneously high-resolution spectral data of the two types of GO suspensions. FTIR spectra showed the oxygen functionalities of the nGO-surface, as well as confirmed the PEGylation process in nGO-PEG nanoparticles suspension (Figure 1A). The FTIR spectrum of GO (Figure 1A) was interpreted with the presence of the oxygen functional groups, such as hydroxyl and ether/epoxy group of sp3 hybridized carbons and the carbonyl and carboxyl functional groups of sp^2^ hybridized carbons. The broadband between 3650–2550 cm^−1^ indicated OH stretching vibrations characteristic of hydroxyl and carboxyl groups and adsorbed water. The high-wavenumber shoulder at 3610 cm^−1^ and the deformation mode observed at 1403 cm^−1^ were assigned to the hydroxyls groups. The band centred at 3430 cm^−1^ together with the bending mode around 1610 cm^−1^ suggested the presence of adsorbed water on the nGO surface. The OH– stretching vibrations, represented as a wide overlapping band centred on 2950 cm^−1^, together with the band of 1725 cm^−1^, could be explained by the presence of carboxyl groups (COOH). The peak at 1725 cm^−1^ was indicative of C=O stretching vibrations of carbonyl groups. The peak at 1285 cm^−1^ was assigned for the C(sp2)–O simple bond from carboxyl and/or carbonyl groups. The presence of ethers was proven by an intense absorption band around 1087 cm^−1^ and a shoulder at 998 cm^−1^, which corresponded to C–O stretching vibrations and the two shoulders at 957 and 840 cm^−1^ were connected with the existence of epoxides. The in-plane vibrations of the –C=C– bonds from hexagonal aromatic rings of the graphene nanosheets were observed as weak peaks in the interval of 1585–1440 cm^−1^ in the FTIR spectrum of PEG-NH_2_ (Figure 1A). There were two weak peaks at 3250 and 3110 cm^−1^ and one at 1641 cm^−1^. The bands at 2880 cm^−1^ represented the extending shakings of CH_2_ and CH_3_ groups and these at 1473, 1410, 1365, and 1350 cm^−1^ suggested their deformation modes. The –C–O–C– functional groups of the polymeric structure were presented by an intensive triplet with a maximum peak at 1114 cm^−1^ (with an overtone at 1970 cm^−1^) together with the bands at 964 and 850 cm^−1^. The FTIR spectrum of nGO-PEG proved the reduction of nGO and PEG-NH_2_ conjugation. Most of the characteristic bands for PEG-NH_2_ were found in the spectrum of nGO-PEG. Because the conjugation is non-stoichiometric, some of the bands were overlapped. The infrared (IR) spectrum of pristine nGO lacked the band at 1725 cm^−1^. This was due to the elimination of the carbonyl function groups most likely because of the establishment of an amide bond (–CO–NH–) in the process of incorporation of PEG-NH_2_ on the nGO surface. Therefore, we assign the PEGylated nGO to the observed low intensive peak at ~2955 cm^−1^ and the intensive peak at 1641 cm^−1^.

The results from the UV/Vis spectroscopy for nGO and nGO-PEG are shown in Figure 1B. Typically, the spectra of nGO prior PEGylation exhibited an absorbance peak at ~230 nm due to π→π* transition of aromatic C–C ring. A weak shoulder peak at ~300 nm was ascribed to the n→π* transitions of C=O bonds, confirming the presence of oxygen-containing functional groups. This shoulder peak was observed also in ultraviolet-visible (UV-VIS) spectra of nGO-PEG. Coating of nGO with PEG led to a slight red-shift of absorbance peak into 233 nm indicating a very slight reduction process. Compared with nGO, nGO-PEG exhibited higher absorbance at a long wavelength (more than 300 nm) (Figure 1A), pointing that the PEG chains of nGO-PEG strengthened the reduction extent of nGO during the process of PEGylation. The reduction was verified by the colour change of nGO suspension from brown to dark brown during PEGylation (Figure 1B, the built-in photo) due to the reduction of nGO under high-temperature conditions.

The Raman spectra of nGO and nGO-PEG with the characteristic peaks have been presented in Figure 1C. Raman spectroscopy is a fast, non-destructive, and high-resolution tool for the characterization of the lattice structure and the electronic, optical, and phonon properties of carbon materials. The positions, line shapes, and intensities of these peaks give information for investigating the structures and electronic properties of graphene-based materials. Carbon materials without a disorder display only one band (G band) at 1582 cm^−1^. When disorders are introduced in materials, a defect-induced band (D band, D for defect) arises at 1350 cm^−1^ as symmetry is broken. Other bands appear or are modified and the G band shifts, broadens, and is overlapped, but yet intense. In our samples, nGO and nGO-PEG, D, D’, and D + G peaks appeared. The D and D + G peaks belong to the forbidden transitions and prove the presence of defects that are characteristic of the graphene oxide-based materials. The G mode is due to the in-plane stretching vibration of hybridized C-sp2 and displays not only aromatic rings of carbon but also others, such as >C=O function. The 2D mode is always allowed and appears as the most intense feature in a perfect single-layer graphene. This band was not well resolved in the spectra of nGO and nGO-PEG as can be seen from Figure 1C. The absence of the 2D band or an extra-wide band also indicates that GO was dominated by the high degree disordered. The strong and broad D band and an I_D_/I_G_ ratio confirmed the lattice distortions and a large amount of sp3 carbon function. However, the presence of an intensive G peak both in the spectrum of nGO and nGO-PEG showed that the hexagonal aromatic rings of the graphene nanosheets were not destroyed. The I_D_/I_G_ ratio (0.927) in the spectrum of nGO-PEG compared to the spectrum of nGO (0.947) suggested that, during the procedure of nGO-PEG synthesis, the aromatic structures have been recovered by repairing defects. In the spectrum of nGO-PEG, we have found that the 2D band was blue-shifted while the D-band was red-shifted and, together with the appearance of the D’ band, proved the increase of nitrogen incorporation in the sample.

#### 3.1.2. Characterization and Nanoparticles Size Measurements of nGO and nGO-PEG

The morphology and the size of the studied nanoparticles: pristine nGO and PEGylated nGO were studied by transmission electron microscopy (TEM: JEM-2100, Japan) and a Zetasizer (Malvern Instrument, Ltd., Worcestershire, UK). The results of these morphology studies are shown in Figure 2. Figure 2A summarizes the results for nGO. The TEM micrograph illustrates thin and transparent GO sheets with relatively large and smooth surfaces. After PEGylation, the surface of the nanosheets appeared rippled, which favored the functionalization of NPs with drugs or other bioactive molecules (Figure 2B: nGO-PEG TEM image). The multiple-layers structure observed in nGO-PEG was explained by the spatial interactions between the PEG molecules, which had grafted onto the edges of the GO layers. Similar morphological changes in GO have been reported by Yuen-Ki Cheong et al. [31] upon functionalization of GO with 4arm-PEG5K-NH_2_ via amide bond formations. The two types of particle size distribution histograms are shown in Figure 2A,B (right panel). Determinations of the hydrodynamic diameter of nGO and nGO-PEG revealed that nGO nanosheets had smaller particle sizes than nGO-PEG indicating that PEGylation increased the size of the GO NPs. The average hydrodynamic diameter of nGO and nGO-PEG in the aqueous suspension was 252.7 nm and 324.6 nm, respectively (Figure 2A,B, right panels). Distribution of particles by size showed that, after PEGylation, the size distribution increased (with about 20%) because the nGO particle size range was from 43.82 nm to 458.7 nm while, for nGO-PEG, it was from 164.2 nm to 615.1 nm. 

#### 3.1.3. Stability and Aggregation Behaviour of nGO and nGO-PEG Dispersion in Aqueous Solution 

Colloidal stability of nGO and nGO-PEG NPs was probed by additional dynamic light scattering (DLS) studies (Figure 3A). The surface charges represented as zeta (ζ-) potential of the nGO and nGO-PEG NPs measured in water was negative for both particle types, indicating the presence of notably carboxylic functions and ensuring water solubility. The ζ-potential value of nGO was −32.9 mV, while nGO-PEG was −21.6 mV. Significant alterations in particle size and zeta potential were detected for nGO for 10 days, which was not detected for nGO-PEG upon 10 days incubation at 4 °C in aqueous solutions, demonstrating high colloidal stability (Figure 3A). The average size of nGO-PEG NPs was 324.6 on the first day and 320.4 nm on the tenth day. The zeta potential kept the values for the PEGylated nGOs stable and around −21.6 mV for both studied time points, thus, suggesting that nGO-PEG NPs were well dispersed and did not show any agglomeration tendency. In contrast, nGO NPs increased their average particle size from 252.7 nm to 296 nm during the studied period and increased the zeta potential from −32.0 to −19.1 mV during the time kinetics, which indicated relatively low colloidal stability (Figure 3A, nGO). The polydispersity index (PDI) also showed a slight increase in the estimated values after PEGylation, namely from 0.265 for nGO to 0.295 for nGO-PEG (Figure 3A). Data in drug delivery applications using nanomaterials as carriers show that a PDI index of 0.3 and below is considered to be acceptable and indicates a homogenous population of NPs [42]. These results were confirmed after incubation of nGO and nGO-PEG NPs in DMEM medium containing 10% FBS for five days with cells. An aggregation of nGO and well-dispersed nGO-PEG NPs during the five days of in vitro cell culturing was observed and illustrated in Figure 3B, where images are taken on the first and fifth days.

#### 3.1.4. Sterility of nGO and nGO-PEG Nanosheets

When synthesized, nanomaterials are easily contaminated with Gram-negative and Gram-positive bacteria that mask the real biological effects of nanomaterials. Therefore, we screened our graphene derivatives for microbial contamination by a quick agar sterility test where nGO and nGO-PEG nanosheets were streaked on three different types of agar plates (Columbia agar with 5% Sheep Blood, MacConkey agar, and Sabouraud Dextrose Agar) and incubated for 72 h at 37 °C, which are conditions allowing bacterial and fungi growth. All of the GO nanosheets and their derivatives were free of bacterial and fungal contamination (Figure 4). Our sterility tests indicated that nGO and nGO-PEG NPs were sterile and apt for safe application under regular culture conditions.

### 3.2. Biological Experiments for Studying nGO and nGO-PEG NPs Biocompatibility and Cytotoxicity 

#### 3.2.1. In Vitro Biological Effects of Non-NIR-Irradiated nGO and nGO-PEG on Low and High Invasive Colon Cancer Cells

To probe the biological effects of nGO and nGO-PEG, we have performed in vitro assays for cell viability and proliferation studies in the presence and absence of the two types of nanoparticles. The concentrations used for the cell viability tests with nGO and nGO-PEG were 5, 10, 20, 50, 100, 200, and 500 µg/mL (f.c. in culture media). Two different types of colon cancer cell cultures were used: Colon26 and HT29. Colon26 is a mouse high invasive colorectal cancer cell line, while HT29 is a human low invasive one. The results from the cytotoxicity effect of nGO and nGO-PEG on Colon26 and HT29 cancer cells were evaluated after 24 h and 72 h incubation with increasing concentrations of nGO and nGO-PEG NPs. As a control, cells were cultured in the absence of NPs. At these preliminary biological experiments, no NIR irradiation was used.

Figure 5 summarizes the obtained results for cellular viability of colon cancer cells treated with increasing concentrations of nGO and nGO-PEG. Figure 5A presents the cellular viability assays with Colon26 cells after 24 h and 72 h of cultivation with nGO and nGO-PEG, while Figure 5B demonstrates the results with HT29 cells. Following the cellular viability results, we have concluded that no concentrations of nGO and nGO-PEG lower and equal to 100 μg/mL significantly affected the viability of Colon26 cells during the first cultivation time, i.e., 24 h as compared to controls. This tendency was kept for the 72 h of Colon26 cultivation with NPs with a single exception: nGO at 100 µg/mL showed a reduction in Colon26 cellular viability. We have assumed that, due to the highest concentration of nGO and its aggregation tendency, the cellular membranes have been disrupted, thus, leading more cells to death. This tendency was conferred by the treatment of Colon26 cells with nGO-PEG. Only 200 and 500 µg/mL of the tested concentrations of nGO-PEG gave a reduction in Colon26 growth and viability, thus, suggesting that the PEGylation of nGO led to little cytotoxic effect on Colon26 cells. Therefore, for Colon26, we have picked the concentration of 100 µg/mL for both nGO and nGO-PEG NPs for the next experiments. In HT29 cells (Figure 5B), only the highest tested concentration of 500 μg/mL of both nGO and nGO-PEG significantly suppressed cell viability. Again, all concentrations lower and equal to 100 µg/mL of the two types of NPs showed little to absent cytotoxicity, thus, suggesting that 100 µg/mL was the most biocompatible for the next experiments. The logic behind picking up this particular concentration was the main aim of our work, i.e., proving the role of the PEGylation of nGO as a smart step for higher biocompatibility of the developed therapeutic approach for colon cancer cells. Therefore, the next experiments aiming at studying the biological activity of nGO and nGO-PEG were done with 100 µg/mL f.c. of nGO and nGO-PEG in combination with NIR activation.

#### 3.2.2. In Vitro Biological Effects of NIR-Irradiated nGO and nGO-PEG

To determine the combined effect of nGO and nGO-PEG NPs with NIR irradiation, Colon26 and HT29 cells were treated with pristine and PEGylated nGOs at a concentration of 100 μg/mL for 24 h and 72 h as described above and were irradiated with an 808 nm (1.5 W/cm^−2^) laser source for 5 min, followed by cell WST-1 assay (Figure 6). Colon26 cells incubated for 24 h with nGO and nGO-PEG showed a decrease in their viability without NIR activation (Figure 6A, black bars, 24 h). In contrast, the treatment with nGO and nGO-PEG with NIR activation showed no effect on cell viability in comparison with the control cells (Figure 6A, red bars, 24 h). This effect was more pronounced in Colon26 cells treated with nGO-PEG and NIR-activated NPs. After 72 h of incubation of Colon26 cells with the two types of NPs, the cell growth decreased in the two experimental conditions (with and without NIR irradiation) (Figure 6A, 72 h). The same effects were demonstrated on the fluorescent micrographs obtained after FDA experiments (Figure 6A, FDA micrographs). NIR activation of nGO and nGO-PEG NPs led to higher biocompatibility of the tested nanomaterials with the studied highly invasive colon cancer cells, leading from a little effect to an absent effect on cellular viability. The last was a guarantee for bio-suitability of the developed combined therapy between PEGylation of nGO and NIR activation for highly invasive colon cancer. The observed effect in the low invasive HT29 cells before and after irradiation was even stronger when compared to the results with Colon26 cells. FDA staining and WST-1 assays for cell viability after 24 h of incubation with nGO and nGO-PEG NIR irradiation increased the number of viable cells (Figure 6B, 24 h), while the older culture after 72 h of cultivation with the two types of NPs showed no inhibition in cell proliferation (Figure 6B, 72 h). These results were confirmed by the fluorescent imaging after FDA staining (Figure 6B, FDA stained micrographs). The morphological changes in Colon26 and HT29 cells were studied by FDA staining after treatment of the two types of colon cancer cells with 100 g/mL of nGO and nGO-PEG treatment with and without NIR irradiation (Figure 6, FDA micrographs). In general, no drastic alterations in Colon26 and HT29 single-cell morphology after the combined NPs and NIR treatment were found. As can be seen from Figure 6A, the control Colon26 cells tend to aggregate and to form typically for the colon cells’ hollow structures (most prominent in FDA micrographs at 72 h of cell growth), while, in nGO and nGO-PEG-treated cells, without and with NIR activation, Colon26 cells look more homogeneously distributed and without any hollow structures. A stronger synergic effect (NPs + NIR) was detected in HT29 cells. Unlike Colon26 cells, HT29 cells generally do not form hollow structures during in vitro cultivation. Instead, they form large aggregates. After 72 h of exposure to nGO and nGO-PEG NPs combined with NIR irradiation, the HT29 cells formed much smaller aggregates. Altogether, these results are another proof of our hypothesis that PEGylation of nGO in combination with NIR irradiation leads to increased biocompatibility, thus, promising novel, smart therapeutic developments for targeted colon cancer approaches.

#### 3.2.3. In Vitro Wound Healing of Colon Cancer Cells: Inhibition of Cell Migration by nGO and nGO-PEG with and without NIR Activation 

Cell migration is involved in several pathological processes such as tumour invasion, neo-angiogenesis, and metastasis. To investigate the effects of nGO, nGO-PEG, with and without NIR activation on cell migration, we performed the in vitro wound-healing assay with the two types of cell lines after incubation with 100 µg/mL of nGO and nGO-PEG. Results are shown in Figure 7. The effects on cancer cell migratory potential of NPs treatment alone and in combination with NIR irradiation were observed after 24, 72, and 96 h of cellular cultivation. The results obtained from the analysis showed that nGO and nGO-PEG NPs inhibited Colon26 cell mobility after 24 h, 72 h, and 96 h of incubation time when compared to the untreated control (Figure 7A). Regarding nGO, the registered suppression was slightly stronger in comparison to nGO-PEG at the long-term treatment periods of 72 h and 96 h, thus, suggesting higher biological compatibility of PEGylated nGO than of pristine nGO. Combining the effect of NPs and NIR irradiation against Colon26 cell proliferation and motility followed the same tendency as the previously mentioned one, with a higher inhibition by nGO and NIR irradiation. The registered effects at 96 h of incubation were almost similar for both NPs. Quantitation of the wound-healing results showed that Colon26 cells demonstrated wound closure with 17%, 82.57%, and 93.03% success while the control cells that were NIR irradiated showed a slight reduction of the observed wound closure with 11.04%, 59.81%, and 89.89% at 24 h, 72 h, and 96 h of cell growth recovery, respectively.

Concerning the HT29 cell line, the pristine nGO completely inhibited cancer cell mobility at all the studied treatment duration periods while nGO-PEG exhibited a slightly weaker inhibitory effect in comparison to nGO (Figure 7B). The combination of both types of nanographene NPs and NIR irradiation showed a stronger reduction of cancer cells motility. The migratory potential of control HT29 cells detected with the wound healing assay showed significantly lower values of 0.72%, 13.99%, and 20.24% of wound healing for the non-irradiated and 6.44%, 7.9%, and 18.34% for the NIR-irradiated control cells. These results show that the low invasive HT29 colon cancer cells are more sensitive to nanographene treatment in combination with PEGylated NPs and NIR concerning their long-term migratory ability.

To the best of our knowledge, the difference in cell proliferation and motility between the two types of colon cancer cell lines could be due to the different invasiveness. The low invasive HT29 demonstrated little recovery, i.e., slower proliferation potential, after the induced destruction of the cell monolayer with and without the application of NPs. This will be a centre of future experiments for studying the intricate molecular mechanisms of this cell migratory inhibition by the applied combination of nGOs, pristine and PEGylated with NIR, and the colon cancer cell characteristics.

## 4. Discussion

This study aims to design optimized smart nanoplatforms for colon cancer treatment based on the PEGylation of graphene oxide nanoparticles activated with near-infrared irradiation. For this purpose, graphene oxide was first PEGylated to improve biocompatibility, stability, and aggregation of graphene oxide nanoparticles in physiological solutions as well as to increase the NIR adsorption capacity of GO. PEGylation was done following a one-step procedure previously described by Chen et al. [41]. This approach was not only fast and simple but also avoided the need for reaction catalyzers as well as for chemical reagents to convert OH groups of nGO into COOH groups needed for activation of PEG, thus, making as-prepared NPs free of chemical waste and impurities. Before PEGylation, GO flakes were first ultra-sonicated to obtain nano-sized GO (nGO) with an average particle size below 300 nm. Sonication decreases the particle size, without considerably changing other physicochemical properties [42,43], and improves particles penetration into cells. The successful entry of nanoparticles (NPs) into cells plays a vital role in prognostic and treatment efficacy. There are several pathways for graphene internalization including direct penetration, phagocytosis, pinocytosis, micropinocytosis, and clathrin/caveolar-mediated endocytosis, as well as several pathways for excretion and clearance of graphene NPs including lysosome secretion, vesicle-related, and non-vesicle-related secretion [28,44]. Besides the NPs size, other morphological parameters such as surface chemistry, shape, and aggregation are reported to significantly affect cellular uptake and excretion of NPs, but these processes are yet unclear and are under intense investigation. Furthermore, nGO dispersions were simultaneously reduced and covalently functionalized with PEG in a water bath at 70 °C for 24 h. Here, we have modified Chen’s protocol decreasing the reduction temperature from 90 °C to 70 °C to avoid a more significant reduction of oxygen functional groups at the surface of GO and, thus, to prevent the formation of unstable colloidal dispersions in aqueous solutions, which would decrease water solubility [45,46,47]. 

The impact of the one-step reduction and PEGylation of GO on the physicochemical properties of as-prepared nGO-PEG was evaluated by several complementary techniques. DLS results were in accordance with other reports [48,49], demonstrating an increase in the average hydrodynamic diameter from 252 nm to 324 nm after a PEGylation suggesting attachment of PEG molecules to nGO. Moreover, an increase in the particle size distribution and the polydispersity index of nGO was found, pointing that nGO-PEG NPs suspension becomes slightly more heterogeneous. Taking into account, however, that the PDI index for both nGO and nGO-PEG is under 0.3, both NPs are considered homogeneous enough and acceptable for use as drug carriers [42].

Spectroscopic characterization of nGO and nGO-PEG by FTIR, UV-Vis, and Raman spectra analysis confirmed the reduction and functionalization of nGO by mPEG-NH_2_. FTIR spectrum of nGO-PEG exhibited typical characteristic bands for both nGO and mPEG-NH_2_ with a lack of a band at 1720 cm^−1^, proving the successful PEGylation of nGO. Similar preservation of the characteristic groups for GO and PEG in GO-PEG have been reported by others [50,51]. UV-Visible spectra of nGO and nGO-PEG indicated a slight reduction of nGO after PEGylation because of the slight red shift of the absorbance peak from 230 nm into 233 nm and an increased absorbance of nGO-PEG at a long wavelength compared to nGO. The last pointed out that the simultaneous functionalization with PEG increased the extent of the reduction reaction and NIR adsorption and is in accordance with other studies, confirming that Chen’s synthesis protocol allowed recovery of the aromatic structure on nGO-PEG. Other studies have also demonstrated the role of PEG in strengthening the reduction extent of nGO [41,52,53,54]. The analysis of Raman spectra of nGO and nGO-PEG confirmed the recovery of the aromatic structure of PEGylated nGO. Raman spectra further proved the increase of nitrogen content in nGO-PEG samples. Our spectroscopic results are in good agreement with TEM morphological observations of nGO and nGO-PEG NPs. The smooth surface of nGO became more wrinkled and more folded after PEGylation, which could be explained by the spatial interactions between the PEG molecules grafted onto the edges of the GO layers as well as darker because of GO reduction. 

Some data show that the stability of nGO is a result of the physicochemical properties of the nanomaterials as well as of the acidity of the surrounding media [55,56,57]. Therefore, different salts could disrupt the colloidal stability of graphene materials [58], thus, alerting GO flakes mobility, and interfering with the dose delivered to cells [59]. PEGylation improved the solubility and stability of nGO dispersions in water, PBS, and cell culture medium [60,61].

The knowledge of how stable are as-prepared nGO and nGO-PEG materials under storage and cell culture conditions is crucial in order to know how long they can be stored before use and to better understand cell behaviour after NPs-treatment. Therefore, we have used DLS to evaluate the stability of nGO and nGO-PEG incubated in the aqueous suspension for ten days under storage conditions (at 4 °C). The zeta potential and average particle size of nGO-PEG dispersion were stable during incubation time unlike nGO values. Furthermore, we have macroscopically observed nGO and nGO-PEG in DMEM containing 10% FBS under three days of cell exposure and found that nGO aggregated while nGO-PEG were well dispersed. The relatively stable nGO-PEG solutions could be attributed to the PEG molecules conjugated at the surface of nGO nanosheets preventing their interaction and aggregation, which was in good agreement with previous data concerning the colloidal stability of PEG-coated NPs [62].

Before all biological investigations, we have tested the sterility of nGO and nGO-PEG nanosheets to exclude the presence of bacteria and fungi that mask the real biological effect of nanomaterials [63,64,65,66]. In a quick agar sterility test, we have established that nGO and nGO-PEG nanosheets are free of bacterial and fungal contamination, confirming their sterility, which allowed us to perform the biological tests. The effect of the combined therapy of nGO and nGO-PEG NPs and NIR irradiation was estimated on two colon cancer cell lines, representing two different subtypes and molecular platforms as well as a different invasion potential. The HT29 (human colon adenocarcinoma) was used as a model of slow migration and invasion cancer whereas Colon26 (mouse colon adenocarcinoma) was our model for fast migrating and invasive cancer. Both et al. have compared the migration and invasion of various colon carcinoma cell lines into Matrigel [67] and have shown that the Colon26 cells were very aggressive with 75.4% and 31.2% of cell migration and invasion versus 6.7% and 1.8%, respectively, for HT29 cells. Our results from the wound closure assay (for control, untreated, and non-irradiated Colon26 and HT29 cells) are following Both et al. results, confirming the higher migratory potential of Colon 26 cells than those of HT29. We have evaluated the NIR off-cytotoxicity of nGO and nGO-PEG after exposure of the cells to a relatively wide range of NPs concentrations from 5 to 500 µg/mL. Taking into account the high importance of NPs concentration for the optimal therapeutic effect, we have applied the previously mentioned concentrations because they are among the most widely used in in vitro assessment of particle cytotoxicity. On the other hand, exposure of cells to ultra-high NP concentrations to ensure cytotoxicity leads to unrealistic results that cannot be extrapolated into the living systems. Moreover, usually the diagnostic and therapeutic interventions require the administration of minimal concentrations [68]. We have found that both NPs, nGO, and nGO-PEG alone (without NIR irradiation) and below 100 µg/mL were noncytotoxic to Colon26 and HT29 colon cancer cells. Although a tendency for a decrease in cell viability was observed with the increasing concentrations of unmodified nGO, the same was not observed for nGO-PEG for which concentrations lower than 100 µg/mL resulted in higher OD values in the WST-1 assay in comparison to the control. The highest concentrations of 200 and 500 µg/mL suppressed cell viability and growth but to a much lower degree compared to nGO. This result supports the use of biocompatible polymers as PEG to enhance the biological effects of developed nanomaterials [33,34,35,36]. Colon26 cells were more sensitive to nGO and especially to nGO-PEG treatment because the 500 µg/mL nGO-PEG NPs induced almost 100% cell death after 72 h of NPs exposure. 

To further determine the in vitro photothermal effect of nGO-PEG, we have irradiated Colon26 and HT29 cells for 5 min in the presence of the NPs using a 1.5 W/cm^2^ NIR laser. Many studies have shown that NIR irradiation alone was insufficient to induce cell death [14,49], but the combination of NPs with NIR irradiation inhibited cell viability. For instance, Costa-Almeida et al. demonstrated that treatment of non-melanoma skin cancer cells with PEGylated and non-PEGylated NPs resulted in a 38% decrease of cell viability after 30 min of NIR irradiation [51]. The authors have recorded a temperature increase up to 47 °C upon NIR irradiation in PEGylated GO-treated samples. The effect of hyperthermia triggers cell death, thus, activating different molecular changes in the cells, like cell membrane permeability, DNA damage, etc. [14,69,70,71]. Other authors have shown that gold nanorod-assembled PEGylated graphene oxide results in similar levels of cytotoxicity (40% decrease in cell viability) upon irradiation with high power (60 W/cm^2^) Xe-lamp light [67]. The effect of combined therapy depends on many factors such as irradiation time, repetitions, and power of the light source. Therefore, further investigations will be necessary to carefully correlate NIR treatment with cancer cell growth in terms of treatment dosimetry before clinical translation. 

What we have observed here was that the NIR irradiation in combination with GOs NPs suppressed or stimulated cell viability and growth and the effect depended on the type of the cells and the type of the NPs. Besides, nGO and nGO-PEG with and without NIR irradiation showed mild morphological changes in both cell types. The effect of NIR combination with NPs on cancer cell migration was evaluated. The results showed that the migration of both types of cells decreased significantly in both NPs-treated groups compared to the untreated control as the effect was very strong in the nGO-treated group. A possible explanation for the absence of a greater thermal effect of laser irradiation and of a synergic effect (NIR + NPs) on cancer cells could be the low elevation of the temperature in the samples. NIR irradiation was performed at room temperature (23 °C), and the temperature of the cell culture media during the laser irradiation increased by 1–2 °C. This temperature is far from the temperatures achieved by hyperthermia (40 °C), triggering apoptosis and, therefore, could not induce significant alteration in cell growth and motility in comparison to non-irradiated samples. Further efforts are needed to optimize the irradiation conditions. Nonetheless, the effects of hyperthermia, and, in particular, their combination with graphene-based nanomaterials, are still far from being fully understood.

## 5. Conclusions

In the current study, we have PEGylated nGO nanoparticles by one-step thermal reduction and chemical functionalization and have found improved water stability, improved dispersibility, lack of aggregation in complete cell culture medium, and increase in NIR adsorption capacity of nGO. PEGylation resulted in multi-layered and wrinkled nGO sheets with an increased average nano-size of 324.6 nm. Increased biocompatibility of nGO-PEG was found in comparison to nGO when low and high invasive colon cancer cells were treated with a different concentration (5–500 μg/mL) of nGO and nGO-PEG nanoparticles. Cell viability and proliferation were studied. The combination of NIR irradiation (1.5 W/cm^-2^) with both types of nGO NPs at a concentration of 100 μg/mL resulted in a slight alteration in cell morphology, preserved viability, and growth compared to untreated control cells. However, a strong synergic effect of nGO and nGO-PEG and NIR irradiation for 5 min with 1.5 W/cm^-2^ laser was detected on wound closure ability of Colon26 and HT29 cells, but especially on nGO-PEG in low invasive HT29 cells. The last factor suggests that the combined NIR with nGO-PEG therapy could be more effective for low invasive HT29 colon cancer cells.

## Figures and Tables

**Figure 1 pharmaceutics-13-00424-f001:**
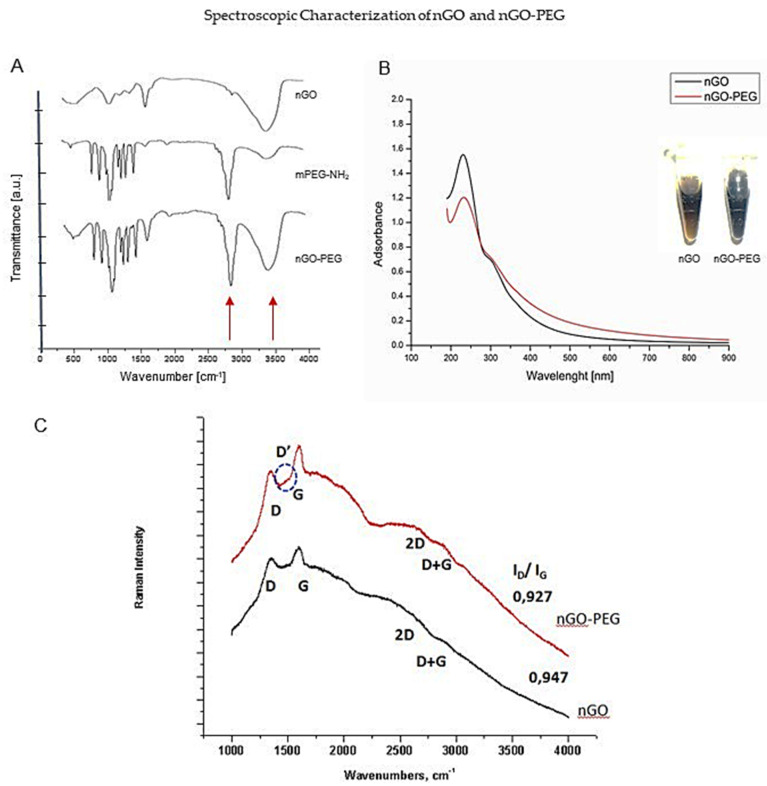
Spectroscopic characterization of nGO and PEGylated nGO (nGO-PEG). (**A**): Fourier Transform Infrared (FTIR) Spectroscopy of nGO and nGO-PEG nanoparticles: Infrared spectra of nGO and nGO-PEG dehydrated samples were recorded using a Thermo Nicolet 6700 spectrometer (Thermo Fisher Scientific, USA) in the mid-infrared region (4000–400 cm^−1^) at a resolution of 2 cm^−1^. The FTIR spectra of Ngo PEG show the proof for the PEGylation modification (red arrows). (**B**): Ultraviolet (UV)-Visible Spectroscopy: UV-VIS-absorption spectra were acquired using a Specord 210 Plus (Edition 2010, Analytik Jena AG, Germany) spectrophotometer. Spectra were recorded in the range of 190–1100 nm. (**C**): Raman spectra of nGO and nGO-PEG with assignment of the important peaks: LabRAM HR Visible Raman spectrometer was used and excitation was performed with 633 nm He-Ne laser.

**Figure 2 pharmaceutics-13-00424-f002:**
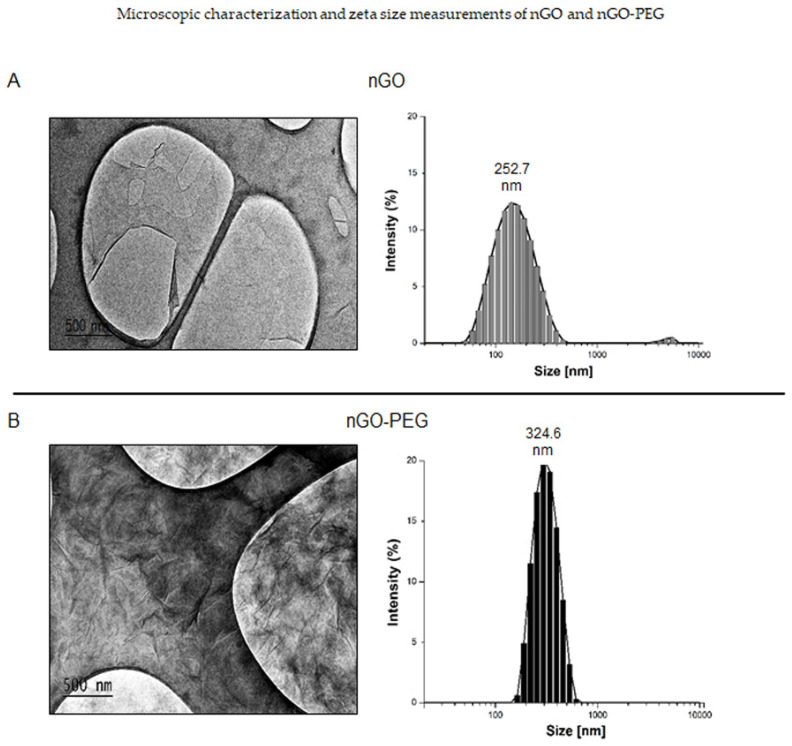
Morphology characterization and particles’ zeta size measurements of nGO and nGO-PEG. (**A**): Size and morphology of the nGO NPs: nGO morphology was studied on JEOL JEM-2100 high-resolution transmission electron microscope (HR-TEM) operated at 200 kV. The nanoparticles’ size was estimated by Zetasizer and results are presented by a histogram on the right panel. (**B**): Size and morphology of the nGO-PEG nanoparticles (NPs): nGO-PEG morphology was studied on JEOL JEM-2100 high-resolution transmission electron microscope (HR-TEM) operated at 200 kV. The nanoparticles’ size are presented by histogram on the right panel.

**Figure 3 pharmaceutics-13-00424-f003:**
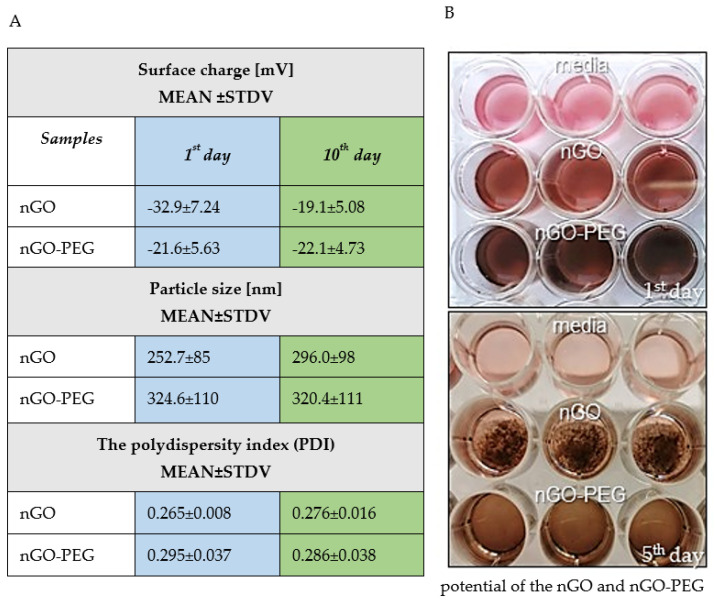
Stability and aggregation behaviour of nGO and nGO-PEG nanoparticles in aqueous and culture media solutions. (**A**) Dynamic light scattering (DLS) experiments for studying the colloidal stability of nGO and nGO-PEG in water: zeta potential, size measurements, and polydispersity index (PDI) of nGO and nGO-PEG for 10 days in aqueous solution. Three independent measurements have been done and the results are represented as MEAN ± STDV. (**B**) Aggregation analyses of nGO and nGO-PEG in complete cell culture media after five days of incubation under optimal culturing conditions. Images are taken on the first and fifth days with Nikon D700 camera.

**Figure 4 pharmaceutics-13-00424-f004:**
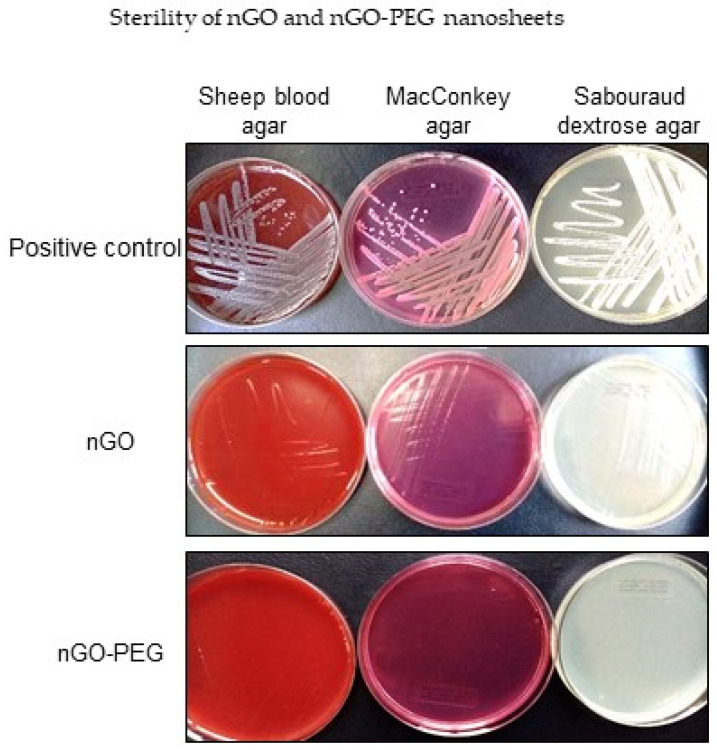
Microbial and fungal sterility of nGO and nGO-PEG nanoparticles suspensions. Pristine GO and nGO-PEG nanosheets were streaked on three different types of agar plates: Columbia agar with 5% Sheep Blood (Becton Dickinson) for Gram (+) and Gram (–) microorganisms; MacConkey agar (Becton Dickinson)—a selective and differentiating agar for Gram (–) bacterial species, and Sabouraud Dextrose Agar (Becton Dickinson)—a nonselective medium for fungi, particularly Staphylococcus aureus, Escherichia coli and Candida albicans were used as positive controls of bacterial and fungal growth. Agar plates were incubated for 72 h at 37°C. Microphotographs were taken with a Nikon D700 reflex camera (Nikon, Tokyo, Japan).

**Figure 5 pharmaceutics-13-00424-f005:**
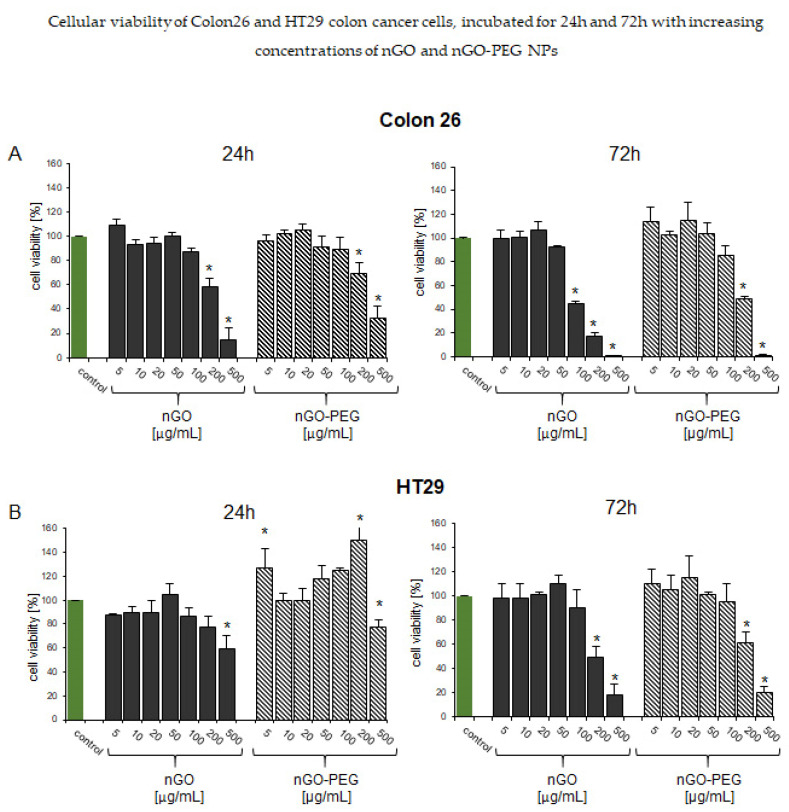
Biocompatibility and cytotoxicity of nGO and nGO-PEG nanoparticles (NPs). (**A**): WST-1 assay after treatment of Colon26 cells with nGO and nGO-PEG with increasing concentrations for 24 h and 72 h. (**B**): WST-1 assay after treatment of HT29 cells with nGO and nGO-PEG with increasing concentrations for 24 h and 72 h. Three repetitions of the experiments are done an values are MEAN±STDV. Statistical analysis is represented as *, where * *p* < 0.05 is accepted as statistically significant.

**Figure 6 pharmaceutics-13-00424-f006:**
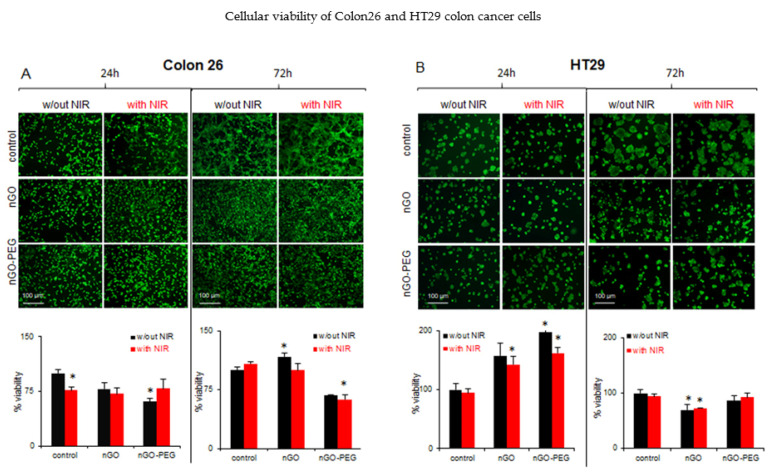
In vitro biological effects of NIR-irradiated nGO and nGO-PEG on low and highly invasive colon cancer cells with and without NIR irradiation. (**A**) FDA staining of Colon26 cells and WST-1 cellular viability assay at 24th and 72nd hour of cultivation with nGO and nGO-PEG. (**B**) FDA staining of HT29 cells and WST-1 cellular viability assay at 24th and 72nd hour of cultivation with the nanoparticles (NPs). Percent viability for the two cell cultures is calculated after three repetitions of the FDA staining and results are MEAN±STDV. * designates statistical significance.

**Figure 7 pharmaceutics-13-00424-f007:**
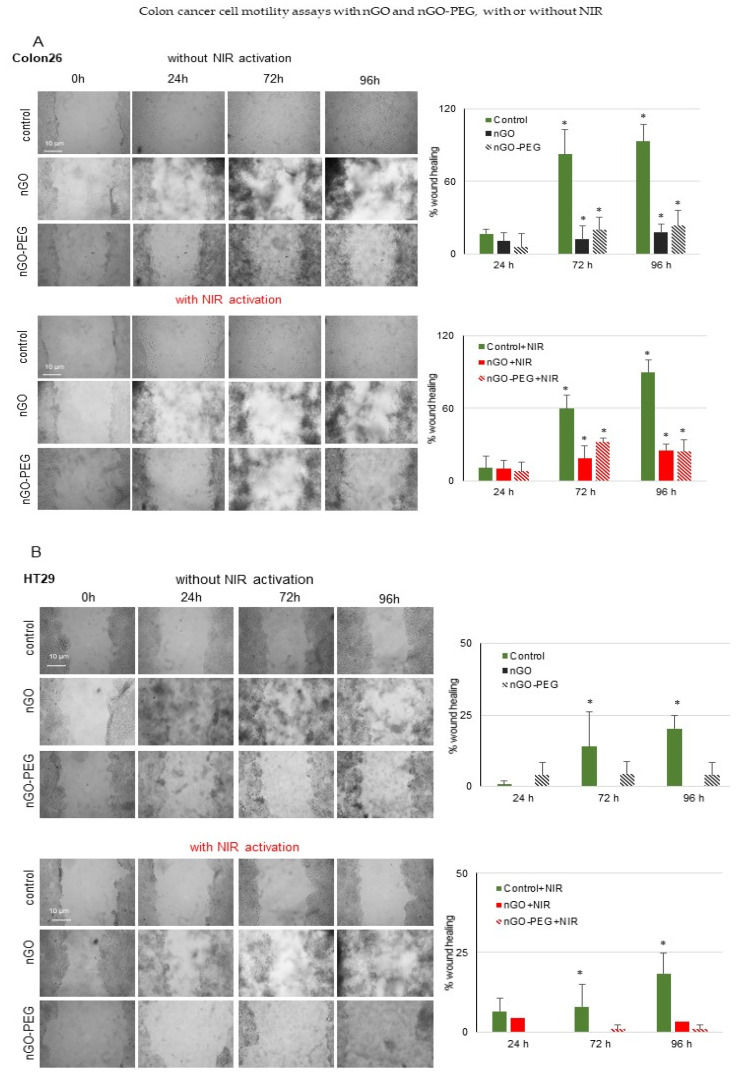
Inhibition of colon cancer cell migration in the presence of nGO and nGO-PEG, with and without NIR activation. (**A**) Wound healing assay of Colon26 cells treated with nGO and nGO-PEG with and without NIR activation. (**B**) Wound healing assay of HT29 cells treated with nGO and nGO-PEG with and without NIR activation. The statistical differences between the control and the treated groups were evaluated by one-way analysis of variance (ANOVA) followed by Dunnett’s post-hoc test and the data are considered significant when *p* < 0.05, designated with *.

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
