# Peer review of "PEGylated Nanographene Oxide in Combination with Near-Infrared Laser Irradiation as a Smart Nanocarrier in Colon Cancer Targeted Therapy"

_pharmaceutics, 2021, doi:10.3390/pharmaceutics13030424_

Round 1
Reviewer 1 Report
pharmaceutics-1135066
General Comments
This manuscript describes the physiochemical properties of polyethylene glycolated nanographene oxide and its therapeutic application to the colon cancer cell lines. The detailed description for the physiochemical properties is interesting, whereas the anti-tumor effect seems to be unclear. Additional experiment would be required to prove the utility of the polyethylene glycolated nanographene oxide.
Major Concerns
- The results of the in vitro studies seem to be preliminary because of the scarce of variety of the cell lines examined. Please add further colon cancer cell lines.
- In the Figure 3A, 5A, 5B, 6A, 6B, 7A and 7B, the number of the sample examined is unclear. Please clarify.
- Significance of statistical analysis in the present study is unclear. In the Figure 5, please declare the control in the figure legend. Conversely, statistical analysis should be performed in the other experiments (Figure 3A, 6 and 7).
- In Figure 3B, the representative images at other different time point, for example Day 0 and Day 10, would be required as a possible control for the aggregation of nGO.
- Figure 5 and 6. the meaning of “K” of remains unclear because of lack of the definition in its figure legends. Please add it.
Minor Points
- Line 274. (Dupan et al. 2011) is unclear. Please explain.
- The list of abbreviations would be required because this manuscript contains the numerous abbreviations.
- This manuscript contains numerous typos. Line 150, 100μg//mL; Line 165, H29;
- Figure 3A should be appeared as a table format.
Author Response
Authors’ replies to comments made by the reviewer:
To whom it may concern,
Dear Sir/Madam,
Thank you very much indeed for all comments made in the manuscript.
We address all of them in the best suitable way.
Please, see below:
Major comments:
- The results of the in vitro studies seem to be preliminary because of the scarce of variety of the cell lines examined. Please add further colon cancer cell lines.
We agree with the reviewer that further studies for the elucidation of the intricate biological mechanisms of the action of nGO and PEGylated GO are needed and they are scheduled in the next steps of our project. The main idea here was to make the first, preliminary steps for the biological characterization of the nanoparticles. Therefore, we have chosen these two types of colon cancer cell lines because they have different invasive potential. The next steps will include deeper analyses of the biological action of the PEGylated GO and a plethora of colon cancer cell lines in order to better study and understand the molecular mechanisms of their action.
- In the Figure 3A, 5A, 5B, 6A, 6B, 7A and 7B, the number of the sample examined is unclear. Please clarify.
All experiments are done in a triplicate way and data are presented as MEAN±STDV. The samples used are GO and PEGylated GO in different concentrations and treated cells from the two cell lines.
Figure 3A is a Table in which all data from the dynamic light scattering (DLS) experiments for studying the colloidal stability of nGO and nGO-PEG in water: zeta potential, size measurements and polydispersity index (PDI) of nGO and nGO-PEG for 10 days in aqueous solution. The number of tested GO and PEGylated GO is 5 in this experiment and is already included as data in the figure caption.
Figure 5A represents the two graphs of the WST-1 assay for the cellular viability after treatment of Colon26 cells with nGO and nGO-PEG with increasing concentrations for 24h and 72h. These data represent the effect on the cellular viability of the nanoparticles without any NIR irradiation. These are preliminary experiments before the activation of the NPs. Three repetitions are done and data are relevantly presented.
Figure 5B represents the same experiments in triplicate for the HT29 cells.
The main idea is to show how increasing concentrations of the nanoparticles will influence the survival of cells without any NIR activation. This allowed choosing the concentration for the two types of NPs, which is active, and at the same timeless toxic to the cells.
Figure 6A FDA staining of Colon26 cells and WST-1 cellular viability assay at 24th and 72nd hour of cultivation with 100 µg/ml nGO and nGO-PEG, with and without NIR irradiation. This concentration was chosen because of the results obtained in Figure 5 A. Again the number of replications of the experiment is three and data are statistically evaluated.
Figure 6B is FDA staining of HT29 cells and WST-1 cellular viability assay at 24th and 72nd hour of cultivation with 100 µg/ml nanoparticles, with and without NIR activation.
Figure 7 represents the wound-healing results from the experiments with Colon26 with and without NIR– Figure 7A, together with data quantitation as graphs. Figure 7B represents the results from the wound-healing assay of the HT29 cells.
- Significance of statistical analysis in the present study is unclear. In the Figure 5, please declare the control in the figure legend. Conversely, statistical analysis should be performed in the other experiments (Figure 3A, 6 and 7).
We have added the statistical significance as * everywhere where needed (in figures 5, 6 and 7). We have corrected the control and named it properly in Figure 5. In Figure 3A the MEAN values ±STDV are included.
- In Figure 3B, the representative images at other different time point, for example Day 0 and Day 10, would be required as a possible control for the aggregation of nGO.
We have added day 1st additionally to Figure 3B. The results from day 10th are the same as in day 5th and we kept these images only.
- Figure 5 and 6 the meaning of “K” of remains unclear because of lack of the definition in its figure legends. Please add it.
We have corrected “K” and named it correctly as control.
Minor remarks:
- Line 274. (Dupan et al. 2011) is unclear. Please explain.
We agree with this comment and have deleted this line.
- The list of abbreviations would be required because this manuscript contains the numerous abbreviations.
We have added a complete list of abbreviations.
- This manuscript contains numerous typos. Line 150, 100μg//mL; Line 165, H29;
Already edited.
- Figure 3A should be appeared as a table format.
We have followed this comment too and have added the table in Figure 3A.

Reviewer 2 Report
This manuscript is a very good one dealing with the functionalization of graphene oxide nanosheets using PEG in order to improve many of its biological and physico-chemical properties.
I recommend its publication after addressing the following:
1- Please comment on the difference in particle size when determined by DLS and by TEM.
2- The combined effect of nanocarriers and near infrared irradiation on the cytotoxicity of cancer cells was previously discussed in: International Journal of Nanomedicine, , ,
Author Response
Authors’ replies to comments made by the reviewer:
To whom it may concern,
Dear Sir/Madam,
Thank you very much indeed for all comments made in the manuscript.
We are addressing all of them in the best possible way.
Please, see below:
1- Please comment on the difference in particle size when determined by DLS and by TEM.
In Figure 2 TEM was used for the study of the morphology of the GO nanosheets and that is we have not discussed this here as no size was measured with this technique.
2- The combined effect of nanocarriers and near infrared irradiation on the cytotoxicity of cancer cells was previously discussed in: International Journal of Nanomedicine, 2020, 15, pp. 2605–2615 and Journal of Drug Delivery Science and Technology, 2018, 47, pp. 176–180
We are grateful for this suggestion and we have cited these works adequately.
Please see lines 60-61 “For this reason, oncological research is investing many efforts towards finding new and efficient therapies that can overcome the limitations of conventional treatments [8-10].”
3- Please determine the nature of the error bars in all the figures captions.
All error bars are the STDV calculated by the programme Excel. Everywhere in the text where necessary, we have added these data in the figures captions.
4- Why haven't the authors mentioned the PDI of the measured nanosheets? I also wonder why the particle size distribution did not seem to be affected by the surface functionalization though usually the distortion of the spherical nature of the particles lead to wide particle size distribution especially in the DLS-intensity based measurements.
We have added the PDI in the table in Figure 3A. The particle size distribution is indeed affected by the functionalization and this is illustrated in text Lines: 351 “Distribution of particles by size showed that after PEGylation the size distribution increased (with about 20%) because nGO particle size range was from 43.82 nm to 458.7 nm while for nGO-PEG was from 164.2 nm to 615.1 nm.”

Reviewer 3 Report
In this submission, Georgieva et al reports the synthesis of PEGylated nano-graphene oxide particles (nGO PEG) along with their combined effect with near-infrared (NIR) irradiation on low and high invasive colorectal carcinoma cells. This is an interesting study. Although there are a few unclear mechanisms which need to be explained. I recommend this paper to be accepted with subject to major revisions. Following are my specific comments;
- Line 68 – it is better to introduce graphene as a sp2 hybridized single-layered material arranged in a honey-combe structure. I suggest the following paper to be cited along with this simple description/definition of graphene (https://doi.org/10.3390/cancers11030319). Authors can add that graphene is found in a wide variety of forms such as graphene oxide, reduced graphene oxide, graphene quantum dots, aerogels etc, where they can highlight that graphene oxide and graphene quantum dots are among the extensively explored derivatives for cancer treatment using light-mediated approaches (photothermal and photodynamic therapy). (https://doi.org/10.2217/nnm-2018-0018) this owes to relatively loo toxicity associated with these two derivatives of graphene.
- Authors have elegantly described graphene oxide and the role of PEG. I suggest to add a little bit more details on why off-target toxicity needs to be overcome and how graphene oxide can induce toxicity, mitochondria respiration, DNA strand break. What are excretion pathways of graphene (oxide)? As such the clearance and excretion mechanisms of graphene is a current area of research activity. Of course it is important to explore this at cellular/subcellular levels to evaluate therapeutic outcome, but it is significantly important to determine where graphene end up at molecular levels. What are potential excretion pathways of graphene. I suggest authors to discuss this around the following paper (1016/j.redox.2017.11.018).
- Line 135 – Zeta should be written as zeta.
- Line 137 degree centigrade needs to be corrected.
- Section 2.5.2 – the toxicity/cell viability has been determined at a very low concentration (different concentrations in a range between 5 and 500 µg/mL). How authors relate this concentration with using GO0functionalized GO) in a living system? Concentration is highly important to achieve optimal therapeutic outcomes with minimal collateral damages. I invite authors to elaborate this in discussion section of the manuscript.
- Raman spectroscopy is very important in the characterization of graphene (oxide and PEGlayated graphene oxide). Can authors add Raman results? If not, can authors describe number of layers of graphene oxide (used in this study). Line 529 – authors stated ‘The layers appeared denser and more folded suggesting an increase in the number of graphene nanosheets’. Is this number referred to number of layers? Please rewrite this sentence, it is confusing in its current format. Again in line 625 – authors stated ‘PEGylation resulted in multi-layered and wrinkled nGO sheets with increased average’. I invite authors to add number of layers. I personally this few-layered is a better word that multi-layered. Multi-layered graphene comes under the category of graphite. As one major difference between graphite and graphene is number of layers.
- Figure 6 – I suggest to add statistical difference in figure legend. It will be very helpful for readers if authors also add that these results are average of three independent experiments. (n=?) This comment applies to all the figures involved error bar (and statistical analysis).
- Figure 7 – I invite authors to add scale bar.
Author Response
Authors’ replies to comments made by the reviewer:
To whom it may concern,
Dear Sir/Madam,
Thank you very much indeed for all comments made in the manuscript.
We address all of them in the best suitable way.
Please, see below:
- Line 68 – it is better to introduce graphene as a sp2 hybridized single-layered material arranged in a honey-combe structure. I suggest the following paper to be cited along with this simple description/definition of graphene (https://doi.org/10.3390/cancers11030319). Authors can add that graphene is found in a wide variety of forms such as graphene oxide, reduced graphene oxide, graphene quantum dots, aerogels etc, where they can highlight that graphene oxide and graphene quantum dots are among the extensively explored derivatives for cancer treatment using light-mediated approaches (photothermal and photodynamic therapy). (https://doi.org/10.2217/nnm-2018-0018) this owes to relatively low toxicity associated with these two derivatives of graphene.
We are grateful for these comments. Therefore, we have corrected and added in the manuscript text the suggested properties in graphene and have cited the suggested papers.
Please, see lines: 78-85 “Graphene is a single-layered material composed of sp2-hybridized carbon atoms arranged in a honeycomb network [18]. Its two-dimensional structure provides a high specific surface area for the binding of a high number of different molecules including drugs, cell-targeting ligands, nucleic acid and proteins efficiently [19-21]. Among the variety of graphene derivates graphene oxide (GO) and graphene quantum dots (GQDs) are the most extensively explored for light-mediated approaches in cancer treatment namely photothermal and photodynamic therapies due to their strong visible and near-field absorbance and low toxicity of GQDs [21-23].”
- Authors have elegantly described graphene oxide and the role of PEG.
I suggest to add a little bit more details on why off-target toxicity needs to be overcome and how graphene oxide can induce toxicity, mitochondria respiration, DNA strand break. What are excretion pathways of graphene (oxide)? As such the clearance and excretion mechanisms of graphene is a current area of research activity. Of course it is important to explore this at cellular/subcellular levels to evaluate therapeutic outcome, but it is significantly important to determine where graphene end up at molecular levels. What are potential excretion pathways of graphene. I suggest authors to discuss this around the following paper (1016/j.redox.2017.11.018).
The comment for the questions why off-target toxicity needs to be overcome and how graphene oxide induce cytotoxicity are explained in Lines 91-103: “The high levels of ROS may cause nucleic acids oxidation, lipid peroxidation, mitochondrial damage, DNA strand-breaks, which result in cell death and may trigger inflammation and fibrosis. However, because the potential anticancer drugs have to be maximally effective in cancer cells with a minimum damage of normal cells to avoid the toxicity of the surrounding healthy tissue, GO NPs need to be improved for their biocompatibility. Graphene-induced toxicity arises from the complex interplay between particle characteristics (e.g. size, shape, surface chemistry and charge, solubility, dispersion, aggregation), particle concentration, and exposure time and cell type. Many studies reported the propensity of GO to form aggregates, particularly under physiological conditions. For instance, fast sedimentation and aggregates formation of graphene nanoparticles inhibit the nutrient uptake of human skin fibroblast cells [30]. Thus, the aggregations and sedimentation of graphene particles compromise their biocompatibility.”
Concerning the second part of the comment, namely the excretion pathways indeed, these mechanisms are yet under investigation. Though following the reviewer comment we have edit the text in the manuscript.
Please see lines: 518-525 “There are several pathways for graphene internalization including direct penetration, phagocytosis, pinocytosis, macropinocytosis and clathrin/caveolar-mediated endocytosis, as well as several pathways for excretion and clearance of graphene NPs including lysosome secretion, vesicle-related secretion, and non-vesicle-related secretion [29, 45]. Besides the NPs size other morphological parameters of NPs such as surface chemistry, shape and aggregation are reported to significantly affect cellular uptake and excretion of NPs but these processes are yet unclear and are under intense investigation.”
- Line 135 – Zeta should be written as zeta.
Yes, we have already corrected it.
- Line 137 degree centigrade needs to be corrected.
Yes, we have already done it. Thank you.
- Section 2.5.2 – the toxicity/cell viability has been determined at a very low concentration (different concentrations in a range between 5 and 500 µg/mL). How authors relate this concentration with using GO (functionalized GO) in a living system? Concentration is highly important to achieve optimal therapeutic outcomes with minimal collateral damages. I invite authors to elaborate this in discussion section of the manuscript.
We agree with the reviewer that NPs concentration is very important for optimal therapeutic effect therefore to evaluate NIR off-cytotoxicity of nGO and nGO-PEG we have used a relatively wide range of concentrations from 5 to 500 µg/ml. In many other in vitro studies the highest tested concentration is lower than 200 µg/ml, therefore we consider that the NPs concentrations used here is very low. Indeed, diagnostic and therapeutic interventions usually only require the administration of minimal concentrations. Therefore, the application of the so-called ‘proof of principle’ approach where cell cultures or experimental animals are exposed to ultra-high NP concentrations to ensure cytotoxicity leads to unrealistic results which cannot be extrapolated into the human scenario.
And we have added this elaboration on the discussed matter in the Discussion section in the manuscript, see Lines: 600-607 “We have evaluated the NIR off-cytotoxicity of nGO and nGO-PEG after exposure of the cells to a relatively wide range of NPs concentrations from 5 to 500 µg/ml. Taking into account the high importance of NPs concentration for the optimal therapeutic effect we have applied mentioned above concentrations because they are among the most widely used in in vitro assessment of particle cytotoxicity. On the other hand, exposure of cells to ultrahigh NP concentrations to ensure cytotoxicity leads to unrealistic results that cannot be extrapolated into the living systems. Moreover, usually diagnostic and therapeutic interventions require the administration of minimal concentrations [69].”
- Raman spectroscopy is very important in the characterization of graphene (oxide and PEGlayated graphene oxide). Can authors add Raman results? If not, can authors describe number of layers of graphene oxide (used in this study). Line 529 – authors stated ‘The layers appeared denser and more folded suggesting an increase in the number of graphene nanosheets’. Is this number referred to number of layers? Please rewrite this sentence, it is confusing in its current format. Again in line 625 – authors stated ‘PEGylation resulted in multi-layered and wrinkled nGO sheets with increased average’. I invite authors to add number of layers. I personally this few-layered is a better word that multi-layered. Multi-layered graphene comes under the category of graphite. As one major difference between graphite and graphene is number of layers.
We are grateful for this comment as it allowed us to look at our results from this point of view and we also performed Raman spectra measurements. The results are presented in Figure 1C in the new version of the manuscript.
What needs to be noted here, though, is the fact that from the Raman spectra, we cannot estimate the number of layers because of the presence of many defects on the surface of two types of nanosheets. This leads to too wide 2D peaks hindering the possibility for precise calculations using ratio I (2D)/ I (G) ratio as well as the full width at half maximum of Lorenzian by deconvolution. Therefore, we have corrected the text cited by the reviewer in lines 529 and 625 in the previous version. Now in the revised one the correction is in lines 554-564: ” Raman spectra of nGO and nGO-PEG agree with Chen et al and Costa-Almeida et al because the lower ID/IG value in nGO-PEG points to a lower degree of structural disorganization and demonstrates restoration of the aromatic structure as a result of reduction and PEGylation of nGO. Also, Raman spectra proved the increase of nitrogen content in nGO-PEG samples as can be concluded from the blue shift of the 2D band and red sift of D’ band. Spectroscopic results are in good agreement with TEM morphological analysis of nGO and nGO-PEG NPs. The smooth surface of nGO became more wrinkled after PEGylation. Pristine nGO-PEG nanosheets appeared darker because of GO reduction and more folded which could be explained by the spatial interactions between the PEG molecules grafted onto the edges of the GO layers.”
- Figure 6 – I suggest to add statistical difference in figure legend. It will be very helpful for readers if authors also add that these results are average of three independent experiments. (n=?) This comment applies to all the figures involved error bar (and statistical analysis).
We have added the statistical difference in the figures captions and in the bars as well in the revised version of the manuscript to all figures involving scale bars. We have further added the number of replicates in all experiments.
- Figure 7 – I invite authors to add scale bar.
We have added scale bars in the figures where necessary.

Round 2
Reviewer 1 Report
The newly preparing revised manuscript is relatively readable because of omitting of the original figures and text. The remaining problems in the manuscript are as follows.
- Fugure 5B. At a glance, the cell viability of 10 and 20μg/mL of the nGO-PEG seems NOT to be significantly reduced in comparison with that of the control (the left green bar).
Possibly, there is statistically significant difference between 5μg/mL and 10/20μg/mL of the nGO-PEG.
Conversely, 500μg/mL of the nGO-PEG seems to be significantly reduced in comparison with that of the control. Please confirm them.
- Figure 6. In the upper FDA staining panels, The abbreviation, K, which means control, still exist.
Please explain it in the figure legends, or replace it with control.
- Generally, each paragraph, especially discussion section, tends to be long, and, therefore, it leads to unclear conclusion.
Please shorten them according to the rule, “one paragraph, one topic”.
Author Response
First, we would like to thank the reviewer for all the critical comments and remarks made.
It indeed increased the quality of our work.
Thank you indeed!
We now lay our replies to the second round of the reviewing process:
"Fugure 5B. At a glance, the cell viability of 10 and 20μg/mL of the nGO-PEG seems NOT to be significantly reduced in comparison with that of the control (the left green bar). Possibly, there is a statistically significant difference between 5μg/mL and 10/20μg/mL of the nGO-PEG. Conversely, 500μg/mL of the nGO-PEG seems to be significantly reduced in comparison with that of the control. Please confirm them."
Yes, you have right the difference is significant for 5 versus 20 ug/ml nGO-PEG. We have corrected the data on the graph.
"Figure 6. In the upper FDA staining panels, The abbreviation, K, which means control, still exist. Please explain it in the figure legends, or replace it with control."
We have already corrected it. It is control.
"Generally, each paragraph, especially the discussion section, tends to be long, and, therefore, it leads to the unclear conclusion. Please shorten them according to the rule, “one paragraph, one topic”."
We have done our best to edit and shorten the Discussion part. All edits are marked by track changes.
Reviewer 3 Report
I am pleased to recommend this paper for publication in Pharmaceutics.
Author Response
We are grateful indeed to all comments made by the reviewer. It increased the quality of our work.
Thank you again!